# Implementation of a Multi-Component School Lunch Environmental Change Intervention to Improve Child Fruit and Vegetable Intake: A Mixed-Methods Study

**DOI:** 10.3390/ijerph17113971

**Published:** 2020-06-03

**Authors:** Nader Hamdi, Brenna Ellison, Jennifer McCaffrey, Jessica Jarick Metcalfe, Ashley Hoffman, Pamela Haywood, Melissa Pflugh Prescott

**Affiliations:** 1Department of Food Science and Human Nutrition, University of Illinois at Urbana-Champaign, Urbana, IL 61801, USA; naderh2@illinois.edu (N.H.); jarick2@illinois.edu (J.J.M.); 2Department of Agricultural and Consumer Economics, University of Illinois at Urbana-Champaign, Urbana, IL 61801, USA; brennae@illinois.edu; 3Office of Extension and Outreach, University of Illinois at Urbana-Champaign, Urbana, IL 61801, USA; jmccaffr@illinois.edu (J.M.); ahoff10@illinois.edu (A.H.); phaywood@illinois.edu (P.H.)

**Keywords:** school meals, nudge, implementation science, food waste, systems

## Abstract

Nudge interventions are widely used to promote health in schools, yet implementation metrics are seldom used to understand intervention outcomes. A multi-component intervention consisting of cafeteria decorations, creative names, social norming taste tests, and flavor station components was implemented in three rural elementary school cafeterias by school nutrition services (SNS) and extension staff. Selection and consumption of fruits and vegetables at lunch were measured through monthly plate waste assessments over eight months (*n* = 1255 trays). Interviews were conducted with SNS staff (*n* = 3) upon completion of the intervention to assess implementation outcomes using validated acceptability and feasibility metrics. Consumption findings were generally inconsistent across schools and time points, yet fruit consumption increased at School 1 (*p* < 0.05) during the taste test and flavor station intervention months and School 2 (*p* < 0.001) during the creative names intervention months compared to baseline. Odds of selecting a vegetable at School 3 were three times higher than baseline during the taste test intervention months (odds ratio (OR), 3.0; 95% confidence interval (CI), 1.3–6.5). Cafeteria decorations and taste tests had higher reported implementation metrics for acceptability and feasibility than other interventions. Thematic analysis underscored the facilitating role of extension support, as well as systems factors, which served as facilitators and barriers across schools and interventions. These findings suggest that nudge interventions are a promising strategy to improve vegetable selection and fruit consumption in school meal programs.

## 1. Introduction

The Dietary Guidelines for Americans (DGA) recommends consuming a variety of vegetables and whole fruits as key components of healthy eating patterns [1], yet less than 10% of children in the U.S. meet the daily fruit and vegetable recommendations [2]. The National School Lunch Program (NSLP), a federally-subsidized program administered by the United States Department of Agriculture (USDA), reaches roughly 32 million U.S. children every day [3]. Children participating in the NSLP can consume up to half of their daily energy intake through school meals [4], but children frequently waste up to 80% of the vegetables served at school lunch [5]. Additionally, children from low-income households and rural settings are more likely to lack access to vegetables outside of school compared to other children [6,7]. Rural schools are often understudied and can face unique challenges to implementing many nutrition and physical activity-related wellness policies [8], including fewer financial resources for school nutrition programs (particularly for small rural schools) [9] and less healthy school food environments [10]. Rural and low-resource schools also have a lower prevalence of salad bar [11] and school garden implementation [12]. Taken together, these findings underscore the need for effective nutrition interventions to promote vegetable consumption in low-income, rural schools.

Many schools across the U.S. utilize low-cost environmental change interventions to promote healthful eating behaviors [13]. Environmental change interventions alter the physical environment to facilitate healthy behaviors [14], and include a subset of strategies called nudge interventions. Nudging is a concept rooted in behavioral economics that involves altering the choice environment to achieve a desired behavior while preserving one’s ability to choose [15] and typically requires very low implementation costs. Generally speaking, nudges are designed to mitigate biases in decision making. For instance, in the case of food choices, consumers often exhibit present-biased preferences, such that they make food selections based on immediate benefits including convenience or food sensory characteristics, over delayed benefits like healthfulness. In the context of school lunch, this is most likely to occur when consumers, or students moving through the lunch service line in this case, are in a “hot” state (which includes experiencing hunger or stress) [16,17]. To combat such biases, nudges work to make healthier options more convenient and appealing to choose [16,17,18,19]. Examples of school-based nudge interventions include making unflavored milk the default option, displaying fruits in attractive ways, and pre-slicing fruits or vegetables to make them easier to eat. Despite the low cost associated with nudge interventions, it is important to understand how feasible these interventions are given the limited staffing and resources available in school nutrition programs at schools in low-income communities [6].

Implementation science (IS) is the study of strategies for adopting, integrating, and delivering evidence-based interventions in diverse settings [20]. IS often considers constructs such as fidelity, feasibility, acceptability, and sustainability to understand intervention outcomes. Research shows strong relationships between implementation strategies and youth health intervention effect sizes [21], and outcomes [22,23,24], yet there remains a critical gap in school nutrition research that incorporates implementation metrics. The primary objective of this study was to understand the effectiveness of a multi-component nudge intervention on fruit and vegetable selection and consumption. The secondary aim was to evaluate implementation metrics of nudge interventions, with a focus on acceptability, feasibility, and fidelity.

## 2. Materials and Methods

### 2.1. Study Design and Setting

The current study used an explanatory sequential mixed-methods design [25] at three rural elementary schools participating in the NSLP in Illinois. It consisted of a non-randomized multicomponent intervention that was evaluated with monthly plate waste data supported by qualitative interviews with school nutrition managers. Schools were recruited to participate in the study through a University of Illinois Cooperative Extension program partnership. Cooperative Extension is the outreach arm of the University located in each county across the state of Illinois and maintains relationships with local entities making it feasible to form community partnerships. Schools 1 and 2 operated a serve-only lunch service (i.e., students have no choice in meal components) and School 3 had offer versus serve (OVS), allowing students to select or decline some meal components. None of the schools in the study had a salad bar. Nudge interventions targeted vegetables (green peas at School 1 and broccoli at Schools 2 and 3) and fruits. Fruit items varied by date within schools.

### 2.2. Participants

Students in kindergarten through 8th grade at Schools 1 and 2, and 2nd–3rd grade at School 3 participated in the study. Students gave verbal assent to participate in the study, and parents were informed of the research by a letter. Participants were selected via assigned ID number labels, which were randomly placed on 40% of trays prior to the start of lunch. On each data collection date, students who received a lunch tray with an ID number on it were invited to participate. School nutrition services (SNS) staff from each school were recruited for interviews following the conclusion of the interventions. SNS staff provided informed consent. The study was approved by the (blinded for review) Institutional Review Board (IRB #19233).

### 2.3. Intervention

The study was conducted over the 2018–2019 school year and consisted of up to four intervention components (cafeteria decorations, creative names, social norming taste tests, and flavor stations). Extension staff collaborated with school nutrition managers to identify nudge and environmental change interventions that would be feasible to implement at their school [26], and assisted SNS staff with the delivery of intervention components. Intervention components and their respective timelines are provided in Figure 1.

#### 2.3.1. Cafeteria Decorations

Decorations were displayed on the walls and along the serving lines in the school cafeterias. Decorations included colorful fruit and vegetable posters, MyPlate decals, and student artwork. At one school, students were involved in creating a large fruit and vegetable rainbow mural displayed in the cafeteria.

#### 2.3.2. Creative Names

Extension staff worked with SNS staff to create creative names for fruits and vegetables served at lunch. One school food service director encouraged students to create some of the names for the food items. Examples include “Brain Boosting Broccoli” and “Power Packed Peas”. Creative name signs were designed to be displayed along the service line in each cafeteria on days when items were offered.

#### 2.3.3. Social Norming Taste Tests

Extension staff and researchers implemented the vegetable social norming taste test component independent of SNS staff. Researchers and extension staff created new recipes for the vegetable components at each school and served them as a taste test during the lunch period. Garlic broccoli or parmesan peas (whichever would be served on the next day’s menu) were offered in small, single serving cups to students the day before each plate waste data collection during the taste test intervention. After tasting the new vegetable recipe, students were allowed to vote on whether they would try the vegetable again. In an effort to provide a social norm nudge to encourage broccoli or pea selection and consumption, a poster with the tallied student taste test vote result was posted in the cafeteria the next day when the tested item was on the school lunch menu, which was also the day plate waste was measured. The vote results were posted each subsequent time the menu item was served.

#### 2.3.4. Flavor Station

A flavor station with spices and seasonings was set up in the cafeterias for student use. Students in all grades were allowed to access the table throughout the lunch period. Flavor station items included lemon pepper, garlic powder, parmesan cheese, and assorted herbs.

### 2.4. Data Collection Methods

#### 2.4.1. Plate Waste

Meal participation and plate waste data were collected once per month at each school, on either Tuesdays or Thursdays. Data were collected during the 2018–2019 school year, from October until May at School 1, from January until April at School 2, and from January until May at School 3. For each school, the same menu items were available on each plate waste assessment date to mitigate the impact of student taste preferences for the entrée served. Baseline data collection was conducted one month prior to the start of the interventions at each school. School meal participation data were obtained from school meal production records at all schools. Students were not informed of when the data collection would take place or that the data collection was related to the interventions. Prior to the start of the lunch period, researchers collected five samples of each menu item to calculate an average weight to act as a standard reference for meal component serving sizes. Students entered the lunch service line using typical procedures and received their trays, which had been randomly labeled with ID numbers prior to the lunch period. As students exited the lunch line, researchers obtained verbal assent and at the OVS school, measured meal component selection using digital photography using Dell Venue 8 model tablets [27].

At the end of the lunch period, students were dismissed from the tables in the cafeteria by grade level and brought their trays to the researchers. Upon receiving each tray from a participant, trained researchers recorded the participant’s grade and tray ID number, and visually assessed the participant’s gender. Meal component waste in grams from each participant’s tray was individually weighed [28] and rounded to the nearest 0.5 g with digital scales (Taylor Professional commercial food scale).

All researchers and extension staff attended a 1.5 h data collection training on the interventions, data collection procedures and forms, and obtaining participant assent. Researchers also received hands-on practice assessing meal component selection and measuring tray waste using tablet cameras and digital scales.

#### 2.4.2. Interviews

Phone interviews were conducted with a nutrition services staff member from each school, lasting approximately 45 min each. Interviews were audiotaped, transcribed verbatim, and assessed for transcription quality. A structured interview protocol was used to assess facilitators and barriers to implementation, cost, and sustainability of school lunchroom interventions. Interview participants were asked questions on feasibility and acceptability for each intervention. These questions were designed to assess evidence based practice implementation, and were previously validated for use in mental health settings [29]. Participants were asked to rate the feasibility and acceptability of each intervention component from 1 to 5 (with 1 corresponding to the lowest rating, and 5 the highest rating) and were asked to provide open ended comments to explain their rating.

#### 2.4.3. Fidelity Observations

Fidelity observations were conducted to assess the extent to which intervention components were delivered by SNS staff as intended by research staff. Twenty-two unannounced fidelity observations were conducted by extension staff throughout the study. Eleven fidelity observations were conducted at School 1, six were conducted at School 2, and five were conducted at School 3.

### 2.5. Data Analysis

#### 2.5.1. Plate Waste Data

All plate waste data were dual-entered by two trained research assistants using a standardized electronic form. The first author then compared their entered data for accuracy and reconciled any discrepancies using the original data collection forms. Quantitative data were analyzed using Stata (15 MP software; College Station, TX, USA) [30]. Descriptive statistics were calculated to characterize the overall study sample. Estimated marginal means for monthly fruit and vegetable selection and consumption adjusted for gender, grade, entrée consumption, and fruit offering. Food selection outcomes were binary and were presented as the percent of participants who selected items from each food group. Waste data were used to calculate percent consumption for each meal component. Logistic regression models (controlling for gender and grade) were used to predict vegetable and fruit meal component selection (compared to baseline) at School 3, which was the only school that allowed students to select their meal components. Multiple linear regression models (controlling for gender, grade, entrée consumption, and meal component reference weight) were conducted to predict the effects of the intervention components on student fruit and vegetable consumption and waste at all schools (compared to baseline). Significance for all analyses was set to *p* < 0.05.

#### 2.5.2. Interview Data

School nutrition service staff interviews were analyzed using ATLAS.ti (Version 8; Berlin, Germany) [31]. Each interview was dual-coded by researchers using a deductive methodology [32]. A codebook was created with existing research questions and additional codes were added as new themes emerged. Interview transcripts were then assessed for thematic trends [33]. Respondent validation was conducted by soliciting feedback from each SNS staff on thematic analyses results and the researchers’ interpretations of the interview transcripts. SNS staff responses to acceptability (*n* = 4) and feasibility (*n* = 4) questions were averaged across each intervention component at each school to create implementation metric ratings for both constructs.

#### 2.5.3. Fidelity Observation Data

Researcher observations from unannounced implementation fidelity checks were used to describe differences in intervention implementation across schools and intervention components.

## 3. Results

Approximately 760 students were exposed to the nudge interventions during the studies between the three schools, with a total of 1255 trays sampled across all data collection periods. Between 40% and 45% of students participating in school lunch at each school were sampled. Across the intervention, 49% of the plate waste assessment participants were female, and 51% were male. Approximately 34% of the plate waste assessment participants were Kindergarten, 1st or 2nd grade students, 38% were 3rd, 4th, or 5th grade students, and 28% of the participants were 6th, 7th, or 8th grade students. Participant and school-level characteristics are summarized in Table 1.

### 3.1. Intervention Implementation and Fidelity

Overall, the intervention fidelity was high, but differences in school infrastructure and resources required different implementation approaches, as outlined in Table 2. The only instance of poor implementation fidelity occurred at School 3, during the creative names intervention. Creative names signs could not be displayed along the service line at School 3 as they would interfere with the SNS staff’s ability to efficiently serve students. Instead, creative names signs were posted on small stands and placed on top of the lunch tables throughout the cafeteria. During unannounced fidelity observations throughout the study, Extension staff found that all other aspects of the interventions were implemented as intended (creative name signs displayed along the service lines in Schools 1 and 2, flavor station set up and maintained throughout the designated intervention months at Schools 1 and 3). In addition to these implementation differences outlined in Table 2, challenges with vegetable preparation at Schools 2 and 3 had to be addressed during the intervention. School 2 initially served broccoli spears, which seemed cumbersome for students to eat. The menu was adjusted, and School 2 agreed to serve broccoli florets at the request of researchers in April. At School 3, the broccoli offered was being prepared and held before service in a manner that compromised the quality of its appearance and texture. SNS staff at School 3 adjusted their lunch preparation protocol at the request of researchers in April to ensure a higher quality of the broccoli being offered.

### 3.2. Plate Waste

Trends over time in percent selection, and estimated marginal means for percent consumption of fruit and vegetable components are shown in Figure 2.

Linear regression results for fruit and vegetable consumption at School 1 are provided in Table 3. Compared to baseline, percent fruit consumption was significantly higher in School 1, particularly throughout the taste test intervention component months, as well as the flavor station component months (*β* = 14.2, *p* < 0.05 in March; *β* = 20.6, *p* < 0.01 in April). This increase of 14.2 percentage points in fruit consumption in March and 20.6 in April corresponds to a reduction of roughly 10.5 and 19.4 g of fruit waste per student on average, respectively, compared to baseline fruit waste levels. Vegetable consumption was significantly lower compared to baseline from January through May, with the lowest percent consumption in January (*β* = −22.4, *p* < 0.001), which corresponds to an increase of 19.0 g of vegetable wasted per student on average. Change in percent vegetable consumption in School 1 during the first taste test month (March) compared to the month prior to taste test implementation was positive (*β* = 3.1, *p* = 0.54), although not statistically significant (data not shown in table).

Linear regression results for fruit and vegetable consumption at School 2 are provided in Table 4. The percent of fruit consumption was significantly higher in School 2 during the first creative names intervention month compared to baseline (*β* = 19.2, *p* < 0.001), while the percent of vegetable consumption was significantly lower during that month compared to baseline (*β* = −20.1, *p* < 0.001). The increase of 19.2 percentage points consumption of fruit in February compared to baseline corresponds to a reduction of roughly 18.9 g wasted per student on average, while the decrease of 20.1 percentage points vegetable consumption corresponds to roughly 15.7 more grams of vegetable wasted on average. The percent of vegetable consumption was significantly higher in School 2 during the first taste test month (March) compared to the month prior to taste test implementation (*β* = 19.3, *p* < 0.001), corresponding to a 15.1 g reduction in vegetable wasted per student on average (data not shown in table).

Logistic regression results for fruit and vegetable selection, and linear regression results for fruit and vegetable consumption at School 3 are provided in Table 5. When all covariates were held constant, the odds of students selecting broccoli in School 3 were roughly 3 times higher compared to baseline during the first two taste test months (March odds ratio (OR), 3.0; 95% confidence interval (CI), 1.3–6.5; April OR, 2.6; 95%CI, 1.1–5.8). Linear regression results for percent vegetable and fruit consumption were not statistically significant, although in months where selection significantly increased (March and April), there was also a decrease in vegetable consumption by 5.8 and 16.9 percentage points compared to baseline, respectively, corresponding to an increase of 4.9 and 10.7 g of broccoli wasted per student on average. The odds of students selecting fruit in School 3 were 9.2 times higher compared to baseline during the last month of the study (May OR, 9.2; 95% CI, 1.1–79.2), when the cafeteria decorations, social norming taste test, and flavor station were all simultaneously implemented. There was no significant difference between fruit consumption at baseline and any of the final three months of the intervention. Change in percent vegetable consumption in School 3 during the first taste test month (March) compared to the month prior to taste test implementation was positive (*β* = 13.0, *p* = 0.29), although not statistically significant (data not shown in table).

### 3.3. Interviews

One SNS staff from each school, who were present on all intervention days, participated in qualitative interviews. Participating SNS staff included one kitchen manager, one head cook, and one food service director. On average, SNS staff had 5.3 years of experience at their respective schools. Two key themes emerged in coding and analyzing interview transcripts: extension support and relationship and influence of systems factors (see Table 6). Interview participants at all three schools viewed their respective Extension staff partners as crucial in supporting the SNS staff throughout the study, particularly in implementing the cafeteria decorations and taste test intervention components. Extension staff supported the schools by providing technical assistance and supplies for all of the intervention components, while the SNS staff were primarily responsible for maintaining the creative names and flavor station components.

The second theme to emerge, the influence of system factors, illustrates school-specific facilitators and barriers to the intervention fidelity and implementation at all three schools. Examples of system factors that were perceived as barriers to implementation by the SNS staff included limited time and other resources available to SNS staff, school staff culture and attitudes, and limited cafeteria space. SNS staff perceived cafeteria space as a barrier particularly for the taste test (School 2) and creative name interventions (School 3). SNS staff resource and time constraints also emerged as perceived barriers to designing and implementing the intervention components at all three schools, but particularly for School 1. In addition to limited resources, SNS staff at School 1 described the attitudes of non-SNS staff at their school (cafeteria monitors and teachers, for example) as resistant to the intervention components, which may reflect the school staff culture and readiness for change as a systems factor relevant to implementation. SNS staff at School 1 and School 2 described the creative names component as requiring some more time, effort, and research to carry out, with SNS staff at School 1 explaining that they needed more support from extension staff to implement the component. In comparison, SNS staff at School 3 emphasized that their cafeteria space and layout was the greatest challenge to implementing the creative names intervention.

Quantitative implementation metric ratings for acceptability and feasibility were collected from SNS staff at all three schools, shown in Table 7. Among all three schools, the lowest average implementation score was 4.25, collected from School 3, which was primarily driven by low acceptability and feasibility metrics for the creative names intervention. Table 7 also outlines illustrative quotes pertaining to the acceptability and feasibility measures assessed in the interviews. The increased effort required to implement intervention components, and the potential for interventions to benefit students, emerged as themes in the SNS staffs’ perceived acceptability of the intervention components at all three schools. For example, SNS staff viewed the cafeteria decorations as highly acceptable, whereas the other components (that may have had more complex or resource-intensive implementation processes, or required more effort to maintain) were perceived as less acceptable. When asked to justify their relatively low ratings of creative names feasibility and acceptability, the SNS staff at School 3 felt this intervention was not appropriate for her school’s student population (ranging from 2nd and 3rd grade), saying:

“I don’t think [the creative names] are doable with that age group. I think it’s confusing to them. I mean… if [Extension staff] wanted to try it, I was going to help them, but in the back of my mind I was thinking: ‘I don’t know whether [the creative names] will work or not’… A lot of our second graders can’t read anyways…. [the students] would say ‘what’s that?’ And [SNS staff] would just say ‘it’s broccoli with a name.’”

## 4. Discussion

To our knowledge, this explanatory sequential mixed-methods study is the first to utilize acceptability and feasibility implementation metrics alongside fidelity measures to better understand the meal component selection, consumption, and waste outcomes of school-based nudge interventions. Mean percent vegetable selection improved throughout the study, with the highest averages occurring over the first two months of the social norming taste test intervention at School 3. The most promising fruit consumption results were observed at School 1 during the cafeteria decorations, social norming taste test, and flavor station intervention components, with significantly higher fruit consumption compared to baseline measures. The increase in fruit consumption observed during the cafeteria decorations component may be, in part, attributable to the level of student involvement in implementation, as increased fruit consumption was also observed at School 2 during the creative names intervention, which also incorporated a high level of student involvement. It is important to acknowledge that while the social norming taste test and flavor station intervention components were specifically targeting vegetables, and not fruit, there were more significant and desirable changes in fruit consumption compared to vegetable consumption in School 1. This difference was unexpected, and considerations such as seasonal differences in student vegetable consumption, or a possible ceiling effect on students’ preferences for and consumption of green peas, are potential issues preventing a more definitive interpretation of this difference in vegetable and fruit outcomes.

Statistically significant improvements in fruit consumption observed during the social norming taste tests may be related to high acceptability and feasibility implementation metric ratings, but they cannot be isolated from creative names or flavor station components due to the overlap in intervention implementation. Taken together, qualitative interview data, differences in implementation fidelity, and implementation metric ratings help to explain the variation in intervention outcomes between schools. For example, the creative names intervention component was given higher acceptability and feasibility metric ratings at School 2 compared to School 3. School 2 also had significantly higher percent fruit consumption during the creative names month compared to baseline, while School 3 had a negligible (and non-significant) improvement in percent fruit consumption during the creative names month (which also included cafeteria decorations). Differences in creative name fidelity between School 2 and School 3 should also be a consideration when interpreting the relationship between these acceptability, feasibility, and fruit consumption outcomes, as creative name fidelity varied widely between the two schools. Key differences in creative names implementation between Schools 2 and 3 are in both content and location. School 2 implemented creative names along the service line, at the students’ point of selection, for daily fruit and vegetable menu items, while the structure of School 3′s cafeteria made it impossible to display creative names along the lunch line, at the point of meal component selection. Instead, School 3 displayed a set of creative names for all menu items on the lunch tables where students were eating, making them irrelevant to participants’ selection decisions.

Selection findings were consistent with other multicomponent nudge interventions utilizing similar components (creative names, and signage or decorations) [34,35]. In School 3, vegetable selection significantly increased relative to baseline when the social norming taste tests were implemented (March and April). Although vegetable consumption did not significantly increase relative to baseline, the trend changed in the desired direction during the taste test months at all schools. While only statistically significant in School 2, percent vegetable consumption improved during the first social norming taste test month compared to the month prior to implementation in all schools. These changes in trends support the likelihood that students tried more of the vegetable relative to months prior to taste test implementation. Repeated exposure interventions have been shown to improve children’s vegetable preferences in low-income elementary schools after 6–8 exposures [36,37]. Given that repeated exposures may be necessary to change behavior, it is possible that significant increases in vegetable consumption relative to baseline may occur if the duration of the intervention and study timeline were longer. Although repeated exposures play an important role in developing healthy dietary behaviors in children, it appears as though a trade-off for increased vegetable selection would be the increased food waste children produce in response to new or unfamiliar food exposures.

Themes from qualitative interview data suggest that extension support and school-level systems factors play a key role in SNS staff perceived acceptability and feasibility, and subsequent intervention fidelity. All fruit and vegetable consumption outcomes from School 3 were not significantly different from baseline measures. It is important to consider that School 3 also had the lowest acceptability and feasibility scores of the three schools and faced particular implementation challenges due to systems factors (for example, cafeteria structure impacting creative names fidelity). In addition, SNS staff from School 3 felt that the appropriateness of the intervention for the younger students at her school was central to the intervention’s acceptability and feasibility, suggesting that appropriateness may be an important implementation metric to include in future studies. Furthermore, it is interesting that creative names received the lowest feasibility ratings of all the studied interventions given this intervention has been among the most studied [35,38,39,40] nudge interventions. The flavor station intervention required more sustained daily effort from SNS staff to set it up and manage the spice inventory. Yet, SNS staff gave it favorable feasibility and acceptability ratings and expressed interest in continuing it beyond the study. This suggests that the type of effort required of SNS staff may impact their feasibility and acceptability ratings. The marketing and promotion tasks required to implement creative names may not be as familiar to SNS staff, whereas the ones required to implement flavor station likely are used daily in other SNS staff responsibilities. Future school nutrition interventions should consider the level of familiarity of the task being requested of SNS staff and assess the need for training and/or positive reinforcement to improve SNS self-efficacy for unfamiliar tasks.

The social norming taste test intervention component had the highest acceptability and feasibility ratings across all three schools. This result was expected as extension staff and researchers implemented this intervention component with little effort required by SNS staff, which was confirmed by the positive feedback regarding extension support provided in the qualitative interviews. Although metric ratings for feasibility and acceptability have not previously been used in school settings, clinical studies have shown that intervention feasibility and acceptability are associated with a higher intervention effect size [41], further underscoring the value of adding evaluative implementation constructs in further studies.

Two of the intervention components implemented in this study address all aspects of the behavioral economics nudge framework: the creative names and social norming taste tests. The cafeteria decorations and flavor station intervention components focus more on augmenting the school cafeteria environment to support healthy choices, whereas creative names and social norming taste tests also address students’ present-biased preferences. For example, a social norming taste test intervention may function mechanistically by shifting the school norm, or default, to tasting broccoli or peas, whereas a flavor station focuses less on convenience and norms, and more on giving students autonomy in seasoning their vegetables. The social norming taste test also specifically draws on the principle of the social norm approach asserting that people are heavily influenced by others’ actions [42], and thus the taste test aims to influence the behaviors of students by describing the behavior of the majority. The distinction between environmental change-oriented interventions and traditional nudges may be important in evaluation. While the difference in environmental change intervention effectiveness cannot be disentangled from nudge intervention effectiveness in the current study due to overlapping intervention components, it is likely that interventions that change the environment and also incorporate changes to the norm or default are more successful in behavior change. As exemplified in this study, the greatest improvement in vegetable selection occurred at the initial implementation of the social norming taste tests in School 3, while the largest gains in fruit consumption were observed during the initial implementation of creative names in School 2, addressing the school norms and students’ present-biased preferences.

Results of this study highlight the importance of seriously considering systems factors in the design and implementation of nudge interventions in school settings, and the relevance of implementation measures in evaluating interventions and programs. For example, the convenience of the served form of the vegetable component (serving broccoli spears instead of florets) and the holding times of the component prior to service were two factors that may have influenced the outcomes of the intervention components in the current study. Addressing these systems factors prior to baseline would have better allowed researchers to isolate the relationships between the interventions and vegetable selection and consumption. Researchers should also examine space constraints and consider their potential to impact intervention fidelity and evaluation. Finally, researchers should make efforts to engage school staff in addition to all nutrition services staff in designing and implementing interventions.

This study had important strengths and limitations that should be considered. A primary limitation of the study design was that it did not include a control group, making it impossible to compare effects of the nudges to outcomes of individuals who did not receive any intervention. Another limitation of the study design was that intervention components overlapped with one another, presenting a challenge in isolating the effects of specific intervention components. Unfortunately, many multicomponent nudge interventions do not use study designs or analyses that allow for the isolation of individual component effects [38,39,43,44,45]. This limitation can result in uncertainty about which interventions are driving strong or meaningful outcomes, and could even result in an ineffective component attenuating the impact of another component implemented simultaneously, as effects can vary greatly by component [46]. This intervention component overlap also limits our ability to examine the interactions of acceptability and feasibility with fidelity, and how these interactions influence plate waste outcomes. Furthermore, all study intervention components were short-term, and future research should investigate the long-term effects of school lunch nudge interventions on student dietary behaviors. Studies should also investigate whether sustained improvements in fruit and vegetable selection (as a result of nudge or environmental change interventions) corresponds to increased consumption over time. School closures due to inclement weather and other uncontrollable delays resulted in changes to the original study timeline. Subsequently, intervention timelines were not identical between schools, making it impossible to compare the results of the interventions by school. Individual-level plate waste was not tracked from month to month, and as a result changes in dietary behavior on an individual student level cannot be assessed.

Despite these limitations, this research contributes to the literature by enhancing our understanding of implementation factors including acceptability, feasibility and fidelity in cafeteria-based nudge interventions, and school meal behaviors in rural elementary school children. A strength of the study is that it utilized validated quantitative implementation metrics in a new practice setting using an interview format that allowed the interviewer to check understanding of the question and the participant an opportunity to provide an open-ended justification of their metric ratings. In addition, the study examined the implementation of nudge interventions in understudied, low-resource, rural schools. Plate waste data was consistently collected on Tuesdays or Thursdays, avoiding variation in meal consumption and waste associated with weekend food insecurity that may affect school meal consumption on Mondays and Fridays [47,48]. All menu offerings except for the fruit component at Schools 1 and 2 were kept consistent throughout the study, and fruit component offerings were subsequently controlled for in the analyses. Additionally, researchers reached their goal of randomly sampling at least 40% of the students participating in school lunch at all three schools.

## 5. Conclusions

Nudge interventions may be a promising strategy to improve child dietary behaviors in school meal programs. The implementation of some nudge interventions was associated with inconsistent improvements in fruit and vegetable consumption. Implementation metric ratings for feasibility and acceptability were consistently higher for cafeteria decorations and social norming taste tests compared to other intervention components. Our study demonstrated that extension support and partnerships are well-received by SNS staff and seem to be an appropriate approach to implement school nutrition nudges. Future studies should utilize a systems approach with strong study designs to examine the long-term isolated effects of these intervention components. Future research is also needed to investigate whether nudges and other environmental change interventions are developmentally appropriate and determine potential trade-offs between increasing vegetable selection and subsequent waste in nudge and repeated exposure interventions. Finally, future research should also examine the interactions between implementation constructs like acceptability, appropriateness, feasibility, and fidelity, and how these interactions influence dietary and waste behavior intervention outcomes.

## Figures and Tables

**Figure 1 ijerph-17-03971-f001:**
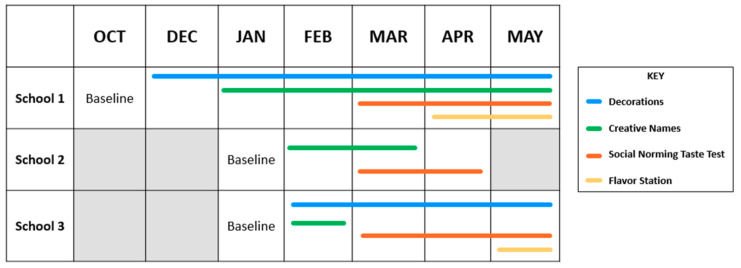
Timeline of intervention components implemented at School 1, 2, and 3. OCT: October; DEC: December; JAN: January; FEB: February; MAR: March; APR: April.

**Figure 2 ijerph-17-03971-f002:**
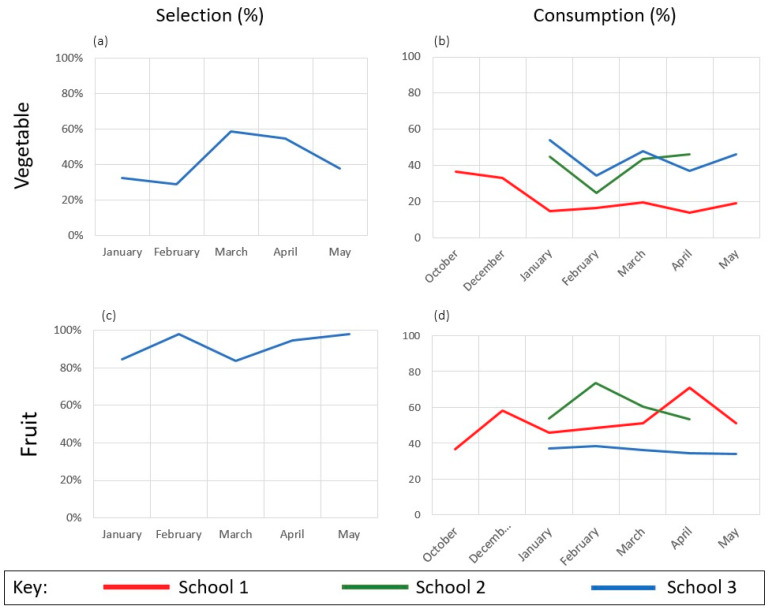
Monthly percent of fruit and vegetable selection and consumption. Note. Original selection frequencies displayed for vegetable (**a**) and fruit (**c**) components (for School 3 only). Estimated marginal means (adjusted for gender, grade, entrée consumption, and fruit offering) displayed for consumption of vegetable (**b**) and fruit (**d**).

**Table 1 ijerph-17-03971-t001:** Demographic characteristics of intervention schools (*n* = 3) and plate waste data participants.

Participant Characteristics			
	School 1	School 2	School 3
Total Number of Trays Sampled	493	483	279
Number of Trays per Month *	61	121	56
Gender	51% Female 49% Male	50% Female 50% Male	44% Female 56% Male
School-Level Characteristics			
	School 1	School 2	School 3
Total Enrollment	188	335	209
Grades	K–8th	K–8th	2nd–3rd
Percent Meal Participation *	84%	82%	62%
Lunch Period Length *	17 min	18 min	34 min
Lunch Service	Serve Only	Serve Only	Offer vs. Serve
Predominant Race/Ethnicity	76% White	88% White	96% White
Free/Reduced Lunch Eligibility	100%	100%	55%

* Numbers displayed are averaged across all months of data collection; min: minutes. K: kindergarten grade level.

**Table 2 ijerph-17-03971-t002:** Variation in intervention process and implementation between all three schools.

	School 1	School 2	School 3
Cafeteria Decorations	Students were involved in creating décor, including a fruit and vegetable rainbow mural.	Not implemented (Cafeteria already decorated prior to study).	Cafeteria was decorated with large nutrition posters, and was re-branded with the school mascot character, as the “Bulldog Diner.”
Creative Names	Names developed collaboratively by SNS and Extension Staff. Creative names displayed along the service line at student’s point of selection daily for fruit and vegetable menu offerings.	Students encouraged to help develop creative name ideas by Food Service Director. Creative names displayed along the service line at student’s point of selection daily for fruit and vegetable menu offerings.	Names developed collaboratively by SNS and Extension Staff. The same set of creative names (for all meal components) displayed on stands on the lunch tables every day, but not at student’s point of selection.
Social Norming Taste Test	Students voted on if they would try the taste test vegetable again by dropping a token into a “Yes” or “No” bucket. Vote results were displayed near the service line the next day.	Students voted on if they would try the taste test vegetable again by raising their hands. Vote results were displayed near the service line the next day.	Students voted on if they would try the taste test vegetable again by dropping a token into a “Yes” or “No” bucket. Vote results were displayed near the service line the next day.
Flavor Station	Flavor station placed against cafeteria wall, away from service line.	Not implemented.	Flavor station placed at the end of one of the lunch tables, near the service line.

**Table 3 ijerph-17-03971-t003:** Intervention component timeline and corresponding regression analyses examining changes in fruit and vegetable consumption over time at School 1.

	December	January	February	March	April	May
Decorations	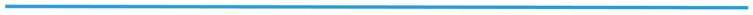
Creative Names		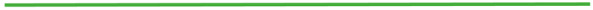
Taste Test				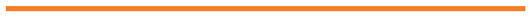
Flavor Station					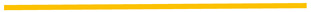
**Consumption**
	***β* (SE)**	***β* (SE)**	***β* (SE)**	***β* (SE)**	***β* (SE)**	***β* (SE)**
Fruit	20.7 (6.4) **	9.1 (6.6)	11.9 (6.4)	14.2 (6.5) *	20.6 (6.4) **	15.3 (8.3)
Vegetable	−4.4 (5.4)	−22.4 (4.6) ***	−21.1 (4.7) ***	−17.9 (4.8) ***	−17.4 (4.8) ***	−16.3 (5.7) **

Note: Linear regression analyses were used for consumption outcomes. Linear regression analyses predicting vegetable outcomes at School 1 used robust standard errors. Variables in the model include: gender, grade, entrée consumption, and fruit offering). Target vegetable at School 1 was green peas. Blue, green, orange, and yellow lines correspond to cafeteria decorations, creative names, taste test, and flavor station, respectively. SE: standard error. * *p* < 0.05, ** *p* < 0.01, *** *p* < 0.001.

**Table 4 ijerph-17-03971-t004:** Intervention component timeline and corresponding regression analyses examining changes in fruit and vegetable consumption over time at School 2.

	February	March	April
Creative Names	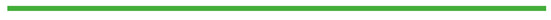	
Taste Test		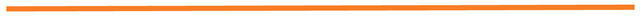
**Consumption**
	***β* (SE)**	***β* (SE)**	***β* (SE)**
Fruit	19.2 (4.9) ***	6.1 (4.8)	−0.4 (4.8)
Vegetable	−20.1 (4.1) ***	−1.2 (4.1)	1.0 (4.0)

Note: Linear regression analyses were used for consumption outcomes. Variables in the model include: gender, grade, entrée consumption, and fruit offering). Target vegetable at School 2 was broccoli. Green and orange lines correspond to creative names and taste test intervention components, respectively. SE: standard error. *** *p* < 0.001.

**Table 5 ijerph-17-03971-t005:** Intervention component timeline and corresponding regression analyses examining changes in fruit and vegetable selection and consumption over time at School 3.

	**February**	**March**	**April**	**May**
Decorations	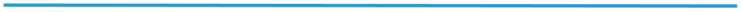
Creative Names	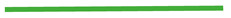			
Taste Test		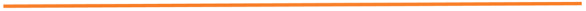
Flavor Station				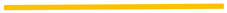
Selection
	OR (95% CI)	OR (95% CI)	OR (95% CI)	OR (95% CI)
Fruit	8.6 (1.0–74.3)	0.8 (0.3–2.4)	2.5 (0.6–10.7)	9.2 (1.1–79.2) *
Vegetable	0.9 (0.4–2.0)	3.0 (1.3–6.5) **	2.6 (1.1–5.8) *	1.3 (0.6–2.9)
Consumption
	*β* (SE)	*β* (SE)	*β* (SE)	*β* (SE)
Fruit	1.0 (8.9)	−1.0 (8.5)	−2.9 (8.8)	−3.5 (8.5)
Vegetable	−19.4 (15.1)	−5.84 (11.8)	−16.9 (12.4)	−7.80 (13.1)

Note: Logistic regression analyses were used for selection outcomes, linear regression analyses were used for consumption and waste outcomes. Variables in the model include: gender, grade, and entrée consumption). Target vegetable at School 3 was broccoli. Blue, green, orange, and yellow lines correspond to cafeteria decorations, creative names, taste test, and flavor station, respectively. SE: standard error. * *p* < 0.05, ** *p* < 0.01

**Table 6 ijerph-17-03971-t006:** Interview themes: Illustrative quotes from school nutrition services staff (*n* = 3) participating in interviews regarding intervention implementation.

Theme	Illustrative Quote
Extension Support and Relationship	“… Well for one, I didn’t know what direction to go in to [decorate the cafeteria], but [Extension staff name] did, and she was very helpful in guiding me in the direction that I needed to go to get [the cafeteria decorations] done … the Extension program seemed to have the staff they needed and the equipment that they needed to get [the study interventions] done, and they seemed organized with what they were doing…” (School 1)
	“We had lots of help from the Extension [staff] … we had enough help, so [the taste tests] went smoothly” (School 2)
	“… Extension did supply the initial cards and everything that was laminated [for the creative names], and they even supplied the stands that I use for them” (School 2)
	“[Extension staff] did a good job of explaining things to me as we went through [the study], and asking my opinion… our opinion matters, and I know [Extension staff] stresses that…” (School 3)
Influence of Systems Factors	“Well, to be honest, the [non-SNS] staff responded negatively because we’re a smaller school. People are not used to change and they felt like, I guess I want to say [non-SNS staff] felt like [Extension staff and researchers] were more in the way of what they do when they’re [in the cafeteria during lunch], trying to take care of the children.” (School 1)
	“We’re in a limited space and our classes are getting bigger and it just, sometimes [the taste test] kind of takes up more room than we have and then it doesn’t seem like the kids have enough time to eat... I mean we’ve got carts and tables around that can be used. It was just finding space to keep [the taste test] out of the way so it wasn’t in the flow with the lunchroom going through” (School 2)
	“We tried to put [creative name signs] out there on the [serving] line… and I couldn’t see the kids because [creative names signs] were just at the right level, so I couldn’t have them there because I couldn’t see the kids with the signs up there… And then I don’t have any other space to put any [creative names signs]. We couldn’t find any other place to put them, so we did not have them through the line” (School 3)

Note: Study interventions included cafeteria decorations, creative names, taste tests, and flavor station.

**Table 7 ijerph-17-03971-t007:** School nutrition services staff quantitative implementation metric ratings and their justification of ratings (*n* = 3).

Intervention	Implementation Outcomes	Implementation Metric Ratings			Illustrative Quotes
		School 1	School 2	School 3	
Cafeteria Decorations	Acceptability *	5		5	“There was nothing to look at but the white walls, and I had asked [Extension Staff name] if she could help me figure out a way to redo it down there [in the cafeteria], so that kids maybe would be more interested in the foods that we serve…I thought [cafeteria decorations] would be a way to change some things up in the cafeteria so that the kids would be more interested and involved in the foods that they eat, instead of just grabbing a tray saying they don’t like it” (School 1) “… I liked them [the cafeteria decorations]. I think they make the cafeteria look bright and colorful” (School 3)
	Feasibility ^‡^	5		5	” …I felt like it was something that we could do, and I didn’t feel like it was going to be a big problem to stress out our staff or our routine” (School 1) “You can just hang up posters. When we put this stuff out it was very easy. There wasn’t any work to it at all, hardly” (School 3)
Creative Names	Acceptability	5	5	2.25	“I figured, why not try it and see if it makes a difference with the kids and the menu?... Because it’s just fun for the kids, and that’s what I’m really here for, is them” (School 1) “I felt [the creative names intervention] was fairly easy to do and like I said, the kids enjoyed it… and I think they tried things normally that they may not have tried just because of the different names” (School 2) “I don’t think I’d ever do it [again]. I’m sorry. Because I don’t like the stands, and I don’t think the creative names help… I just don’t think [creative names] work, and I think [creative names are] confusing to the kids” (School 3)
	Feasibility	4.5	5	2.25	“I had to have help from [Extension Staff]. It’s kind of hard to come up with some of the names that you’re able to put on a menu, and you have to make sure everything is kid-friendly. So that was a little bit more research, but it still wasn’t that big of a deal” (School 1) “[It was] fairly easy to make up the names and find pictures to put with the name itself, and then just make the cards. Was very easy and inexpensive” (School 2)
Social Norming Taste Test	Acceptability	5	5	4.75	“… if [Taste tests] can give kids a chance to try something that they haven’t before, or would not [have tried] before, [then I] was going to give [the kids] a chance to do that” (School 1) “I would do it again, like I said. Just maybe with the different types of [foods] …I just think the kids really enjoyed it too. It was pretty easy to do” (School 2)
	Feasibility	5	5	5	“I think when [Extension staff] explained [the taste test intervention] to me, it seemed like it would work with the kids and… I think it was set up fairly well. [Extension staff] had everything [for the taste test] ready to go.” (School 2)
Flavor Station	Acceptability	5		4.75	“… especially with the older group [of students], [the flavor station] makes them feel like they’re being able to do something that contributes to their meal, and that way they probably would prefer to eat it” (School 1) “I was a little apprehensive because of the age group, but I wasn’t sure how that flavor station would do. Like on the shakers and all, if they would be able to know what they were shaking out, the spices, and if they would do… I think the kids like it, and it works real well. We still have it, we’re going to keep it” (School 3)
	Feasibility	4.5		5	“We came up with a plan to [allow students to use the flavor station] table by table, and if they wanted to they could go up there and get it, and it wasn’t a bunch of people rushing up there. We just had to come up with a game plan as to how we were going to let them go up [to the flavor station]… there was no reason to not try something new just because it might [require] more equipment, or might be more of a mess because... There wasn’t a reason why it couldn’t be done” (School 1)

Note. * Satisfaction with elements of the intervention components. ^‡^ Extent to which the intervention components can be carried out in the specific setting. Implementation metric ratings ranged from 1 (low) to 5 (high).

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
