# Peer review of "Implementation of a Multi-Component School Lunch Environmental Change Intervention to Improve Child Fruit and Vegetable Intake: A Mixed-Methods Study"

_ijerph, 2020, doi:10.3390/ijerph17113971_

Round 1

Reviewer 1 Report

This is a fantastic piece with interesting findings and implications. My comments revolve around improving the paper, especially its linkages to other research in the area of social marketing, improving its background, justification and discussion.

(1) My main concern is that you describe nudging but the interventions are NOT nudging approaches from my experience or expertise - most to me would be classified as social marketing techniques/tools. I would suggest a careful thought or at least acknowledgement about this classification. I also believe the social marketing and consumer behaviour literature can aid your discussion as well as explanation WHY certain interventions were chosen (see my comment below). For example, your social norm approach is well documented as are brand names. see  -

Burchell, K., Rettie, R., & Patel, K. (2013). Marketing social norms: social marketing and the ‘social norm approach’. Journal of Consumer Behaviour, 12(1), 1-9.

Levin, A. M., & Levin, I. P. (2010). Packaging of healthy and unhealthy food products for children and parents: the relative influence of licensed characters and brand names. Journal of Consumer Behaviour, 9(5), 393-402.

In this vein, there is no background provided for each intervention. Why was each intervention chosen? What's your justification?

(2) Another concern is the simultaneous use of interventions (as pointed out in the limitations). The overlaps are also inconsistent between schools (i.e, timing of interventions differ). You may wish to expand on this in your limitations further.

(3) Presenting findings of only three interviews is a long stretch for any research in any publication. Please justify.

Minor issues:

"resource settings can face unique challenges to implementing many nutrition and physical activity-42 related wellness policies (8)," - what are these challenges, please clarify and state.

Otherwise, this is a well written piece with interesting findings!

Reviewer 2 Report

This is a generally well written paper describing a very complex study design in a naturalistic setting of three rural elementary schools.  I have a few comments, mainly related to the positioning of the study and the emphasis the authors put on the different results.

  1. The title cites reducing food waste as if it were a major study theme.  In my reading of the paper, it seems that food waste was simply a measure used to estimate food intake.  Furthermore, there is little mention of the food waste findings in other parts of the manuscript (e.g., data not shown in tables, not mentioned in the discussion, etc.). Other literature might highlight food waste as, for example,  an environmental concern (fueling methane production in landfills).  If its main purpose in this study was to estimate intake, the authors should consider removing it from the title to avoid confusion. 
  2. The actual fruit and vegetable consumption findings reported were quite inconsistent across schools and time frames (were largely non-significant and sometimes, were in the opposite direction of what might have been expected...e.g., vegetable intake decreased during the intervention, plate waste increased), leaving one with the impression that the interventions were not particularly effective at increasing F&V intake.  This contrasts with the findings from the implementation science battery of evaluations...which seemed consistent across the schools.  Perhaps, the implementation science outcomes should be presented as the main findings, while the F&V selection and consumption findings can be presented as secondary.  The authors did a nice job of explaining many of the reasons why the complete battery of interventions was not implemented uniformly across schools and calendar times, and that is likely responsible, at least in part, for the inconsistent findings on F&V intake. However, this does not overcome the observation that there were few significant improvements in actual F&V intake during the study.  
  3. In light of the comments in point 2, the authors should consider revising the abstract, discussion and conclusion sections to emphasize the consistent implementation findings while downplaying the few selected instances where a positive improvement in intake of fruits or vegetables was observed.  
  4. The authors should consider doing a more systematic discussion of what intervention component(s) were associated with increased/decreased F&V intake (in those instances), and why they think that was observed. Although the global findings were inconsistent, there may be instances of interventions having a positive effect that could provide useful insight.
  5. In figure 2, the lower panel shows that in school 3 fruit selection was high throughout the study, but consumption was low (even lower than the other schools).  What do the authors make of this observation?  In the upper panel of the same figure, school 3 shows an increase in vegetable selection, but not for intake.  This raises a larger point that could be brought out in the manuscript concerning how selection is related to intake.  Obviously, selection is on the path to intake, but how how reliable is it?  Do the data in this paper showing increased selection of some items at some times provide encouragement?  Is there literature that shows that consistent selection over time eventually leads to greater intake?  

Reviewer 3 Report

This study has the goal to implement a school lunch nudge intervention aiming to improve the vegetable intake. Variety in food consumption, rich in fibers and low fat, is important for the development of the healthy individual, which indicates the relevance of developing tools that can attract interest for certain foods. Although, I have some questions; authors mentioned in the introduction, line 46, that many schools utilize low-cost “nudging” interventions, but would be appreciable if the authors explain what is considered a low-cost "nudging". Comparing with the current study, the nudge intervention adopted would be considered as high-cost? In the method, line 88, 40% of trays were labeled and students were invited. It is not clean why 40%. People from different countries have different food cultures, which may influence the food choice. Did authors analyze this parameter? Overall, improving vegetable consumption is important in the daily life, providing nutrients and improving the gastrointestinal tract. Maybe for a future research should be investigated different ways of preparing some vegetables, such as steamed and non-steamed.

Round 2

Reviewer 2 Report

The authors have done a fine job of revising the manuscript and balancing the results of the intervention focused on implementation and F&V intake outcomes.  While the results for improving F&V consumption may not have been consistent, this study is useful for highlighting the realities of implementing change in a school setting, identifying essential champions and partners and the importance of involving the students as stakeholders.  

The authors have satisfactorily addressed my concerns/suggestions.